# One-Shot Simple Pattern Detection without Pre-Training and Gradient-Based Strategy

**DOI:** 10.3390/s23229188

**Published:** 2023-11-15

**Authors:** Jun Su, Wei He, Yingguan Wang, Runze Ma

**Affiliations:** Shanghai Institute of Microsystem and Information Technology, Chinese Academy of Sciences, University of Chinese Academy of Sciences, Shanghai 200050, China; j_su@mail.sim.ac.cn (J.S.);

**Keywords:** neural network, machine learning, correlation coefficient, one-shot, bionic

## Abstract

One-shot object detection has been a highly demanded yet challenging task since the early age of convolutional neural networks (CNNs). For some newly started projects, a handy network that can learn the target’s pattern using a single picture and automatically decide its architecture is needed. To specifically address a scenario in which a single or multiple targets are standing in relatively stable circumstances with hardly any training data, where the rough location of the target is required, we propose a one-shot simple target detection model that focuses on two main tasks: (1) deciding if the target is in the testing image, and (2) if yes, outputting the target’s location in the image. This model requires no pre-training and decides its architecture automatically; therefore, it could be applied to a newly started target detection project with unconventionally simple targets and few training examples. We also propose an architecture with a non-training parameter-gaining strategy and correlation coefficient-based feedforward and activation functions, as well as easy interpretability, which might provide a perspective on studies in neural networks. We tested this design on the data we collected in our project, the Brown–Yosemite dataset and part of the Mnist dataset. It successfully returned the target area in our project and obtained an IOU of up to 87.04%, reached 80.28% accuracy on the Brown–Yosemite dataset with disposable networks, and obtained an accuracy of up to 89.4% on part of the Mnist dataset in the detection task.

## 1. Introduction

One-shot or few-shot object detection (OSD and FSD, respectively) and localization have been widely demanded in visual-related projects. Specifically, in our project, which has hardly any training examples, in a scenario where a specified target is set to mark terrains, a model that can not only perform one-shot object detection on simple targets but can also output their location is needed. It is natural for humans to learn a new pattern from a single image and then recognize and point out where this pattern occurs in another picture, so we expect our artificial neural network to fulfill the same task.

In this paper, we propose a model that could learn a simple pattern using a single image, automatically construct a network according to the complexity of the pattern without hand-designed architecture, detect if the learned pattern is present in the testing picture, and if yes, output its rough location.

There are many mature approaches to object detection. Faster R-CNNs [1] and Mask R-CNNs [2], for example, have significant positions in this field. Traditionally, the parameters of CNNs or non-convolutional neural networks are trained by a mass of training data (e.g., thousands of pictures per class) using gradient descent and back-propagation-based methods, such as the classic LeNet [3]. AlexNet [4], VGGNet [5], and ResNet [6] implement data augmentation to alleviate the conflict between the fierce requirements of training data and the limited data supplies. Although data augmentation techniques greatly reduce the training examples needed to train the networks, they are a long way from one-shot or few-shot recognition.

A variety of models have been promoted to address the OSD or FSD problem. The Yolo series [7,8,9], for example, are widely used real-time models. Generally speaking, there are three main forms of access [10] to OSD or FSD: non-episodic-based methods, meta-learning-based methods, and metric-learning-based methods. All three methods use CNNs or non-convolutional networks as the basic module that deals with the image directly or “translates” the image into vectors or metrics. Although they achieve OSD or FSD in the testing phase, their training processes require a relatively large dataset and a large amount of computing. They also contain the “domain gap”/”cross-domain”/”task distance” [11] issue, which may downgrade their performance. 

In a newly started project with simple but unconventional targets to detect and locate, it is neither cost-effective nor practical to collect a big dataset related to the targets or use a very large-scale, well-trained network. Therefore, we need to find a model that can quickly obtain its parameters from a single training sample without any pre-training.

The gradient descent training strategy applied in most neural networks updates the parameters with a speed related to their learning rate. This mechanism requires multiple back-propagations before convergence, which partially accounts for the high training cost. Therefore, we made an attempt to train a network in another way, without gradient-based methods. Since the detection could be carried out by comparing the distance between the training sample and the testing samples, as the metric-based methods suggest, we could directly use the pixel values of the training sample as the template for comparison. To use these pixel values as network parameters, we chose the correlation coefficient as the feedforward and activation function. Accordingly, we created a strategy that would allow the network to deal with the spatial relationship of features directly rather than using convolutional methods. We also designed a process that builds the latter layers according to the complexity of the former layers to enable the network to focus more on features or patterns that require more attention and resources.

In this paper, we propose a model based on a correlation coefficient that gains its parameters directly from the training sample rather than through a training procedure. This mechanism makes it easier to back-trace the activated features and thus makes positioning a simple process. To the best of our knowledge, this is the first time that the correlation coefficient is applied as the activation function to a non-training parameter-gaining strategy that gains the parameters directly from training data in neural networks, which is also a first attempt as far as we know. We also create a construction algorithm that decides the network’s layer number and structure according to the area and complexity of the learned object, making the construction process easier and cheaper. As every activated feature is traceable, the network is also easily interpretable.

## 2. Related Works

Machine learning usually requires a lot of supervised training samples [12]. To satisfy the demands of OSD and FSD and to make more “intelligent” tools that can achieve a performance closer to that of humans, researchers have made great efforts in recent years. Wenbin Li [10] divides the few-shot learning methods into approximately three types: (1) non-episodic-based methods, carried out based on a typical transfer learning procedure. Baseline and baseline++ [13] are typical models of this type. They are first pre-trained on a large dataset with plenty of labeled samples and then test-tuned on a novel dataset with only a few labeled samples for each class. (2) Meta-learning-based methods are designed with a concept of “learn to learn” that imitates human learning methods. Model-Agnostic Meta-Learning (MAML) [14] is a representative work. Such approaches require lots of tasks (conceptually tens of thousands, as suggested by [10]) in the training phase. (3) Metric-learning-based methods usually “translate” the images into vectors or metrics and compute the distances between the testing images and the representative metric of a category. The methods presented in [15,16,17,18] are good examples of this approach. Metric-based methods are simple and fast during testing tasks [11], but the training process for the embedding function can be expensive (it typically requires tens of thousands of training examples and hundreds of thousands after data augmentation [17]).

It could be inferred that all three of these approaches, representing most of the OSD and FSD models, rely on relatively large datasets or a large number of tasks being generated from these training examples. Relying too much on the information coming from the source domain may downgrade their performance if the examples in the target domain are distant or have a different distribution from those in the source domain.

Our main goal is to elide the pre-training progress based on mass data and heavy computation. Giving up pre-training and depending only on the query set in the target domain leads to challenges in fast feature extraction and convergence. This problem can be divided into approximately two parts. The first part, as described in many one-shot or few-shot works, relies on experiences that come from not only the target domain but also many other training examples. For example, given a picture of the front side of a lion, we, as humans, are naturally able to recognize a lion in a picture from the back, even if we have not seen the back of a lion before, and even though the features of the back of the lion are not shown in the frontal picture. This inevitably requires more information than is provided by the only training example, if not pre-programmed into the network. The second part, however, is much easier, as it only involves features shown in the support examples from the target domain. It tolerates a certain level of feature transformation, where features are combined into larger features. In this way, the similarity-comparing strategy is a good choice.

Thus, our design pays more attention to the second part of the problem, and the knowledge-accumulating part is left for future work. In order to cut off the mass data-based training process, which is partially caused by the gradient descent back-propagation strategy in existing OSD networks, we applied the MP neuron model [19] and Pearson correlation coefficient [20]. Using the similarity-comparing idea, we designed a non-training parameter-gaining strategy that allows the network to obtain the parameters directly from the single training example without the gradient descent training process. 

Adam Gaier proved that the architecture of a network may play a more important role than the weight [21], so we focused more on the spatial combination of the activated nodes than their numerical values. This location-related feeding-forward process involves bottom-up and top-down attention logic that resembles human logic [22]. The passive information delivery from former layers resembles bottom-up attention, while the active selection of the needed pattern from a mass of features resembles the top-down process. 

Kanezaki proposes an unsupervised segmentation network in [23]. It first pre-segments the pictures with simple linear iterative clustering (SLIC) and then assigns and merges them with semantic labels. Reference [24] assigns pixels to each category by calculating the probability using the feature distances. Siamese Mask R-CNNs [25] achieved an outstanding performance in one-shot segmentation. In contrast with the Mask R-CNN, it uses a Siamese network to calculate the features of both the reference and testing images.

Without the help of CNNs, the positioning and segmentation of the detected object are accomplished by a back-tracing procedure in our design. This measure is inspired by the method described in [26] that in traditional CNNs, the contribution to the output category could be traced back to every feature and visualized by de-convolution, which also provides an approach to the interpretability of neural networks. In our design, this back-tracing procedure could be even simpler, as the nodes only have two statuses, on and off, making it easy to trace the nodes that activate a pattern. Each node that represents a feature (mostly not formatted) is given a unique ID resembling those in NEAT [27], so they can be easily distinguished from each other. As suggested in [28], constraints such as connectivity could be applied to evaluate images; we also introduce some restrictive checks to filter out unqualified or misactivated features. This back-tracing method reflects the location of the detected targets in a style resembling pixel segmentation, as it can be traced back to the nodes activated in the first layer, which are the smallest feature units in our design.

## 3. Method

The main point of our design is to elide the pre-training process and replace the gradient descent training process, based on the mass data in existing CNNs or non-convolutional networks, with a non-training parameter-gaining strategy. In this way, the network could obtain its parameters without pre-training on large datasets. 

As mentioned in Section 2 on related works, the parameter-gaining strategy in our design is based on the MP model and Pearson correlation coefficient. The MP model, which was proposed by Warren McCulloch and Walter Pitts at the very infancy of neural networks, is a foundational work of artificial neuron networks. It imitates the “all or none” character of biological neurons and is activated when the stimulation reaches the threshold; thus, it works well with the “similar enough, then activate” idea, which is also applied in metric-based methods. The classic MP model, however, only accepts inputs of the Boolean type, which limits the effectiveness of the Pearson correlation coefficient. Therefore, our model accepts an analog quantity for computing the correlation coefficient. In contrast to existing artificial neurons, which also output analog data, it keeps the output Boolean type. This method also directly enables activation sparsity.

We employed the Pearson correlation coefficient as the activation function that enables the network to calculate similarities without a complex embedding function. With this activation function, we could directly obtain parameters from training pictures rather than the gradient descent training process, which is the main training strategy of CNNs and other non-convolutional neural networks. The pattern is copied from the training sample for comparison with the testing sample. We adopted the 3 × 3 sized kernel due to its computational efficiency and good performance, as suggested by [5]. 

The fast construction of our architecture is based on the correlation coefficient weight-gaining strategy, which relies on the design of the network structure. The functional nodes in the network could be divided into two main categories: kernels in the first layer, which capture the features reflected by pixels, and segments in the higher layers, which deal with locational combinations of the nodes in the former layers. Whether a node (which could either be a kernel or a segment) is activated or not reflects whether the corresponding feature is detected in the picture.

The simple target-detecting neural network we propose in this paper is built by a net constructor in the training phase (Figure 1), with a single training picture piled up with an efficient mask. When the training sample is taken into the system, the net constructor builds the network with the input sample. After the network is built, it works independently on testing pictures in the testing phase and locates the object if it exists. The area that the target (or any other activated features) covers could also be provided by the network. 

Later in this paper, we refer to the training process on mass data that helps to build the feature-extractor/learner/embedding function as “pre-training” for convenience. As our design requires no pre-training, we use the only reference picture as our training dataset to obtain all the parameters. In this paper, the training process is the building process, as the network is directly built on the training picture.

### 3.1. Network Construction

The network in our design is automatically constructed by the network constructor (Figure 2). A grayscale picture (the training example itself) is required, an efficient map is also needed to show the area of the foreground that we want the network to learn. 

The constructor will first create a gradient picture that is piled with the grayscale picture and the efficient map. This pile of three pictures will be sent into the kernel maker, which extracts features with a series of 3 × 3 kernels. The segment maker then combines the kernels into bigger segments and builds new layers from these segments. The segment maker recurrently combines these kernels and segments (including those generated from kernels/segments in former layers) until all the segments converge into a single segment that occupies a whole layer. For convenience, we call this node the final node. An end node is produced to represent the class of the training picture. The last connection between the final node and the end node will be established to assign the structure (or sub-structure if multiple examples are learned) built from this example to the category.

#### 3.1.1. Gradient Maker

The gradient maker generates the gradient picture according to the input picture. The gradient value is calculated by Formula (1).
(1)$$p_{i,j}=\sqrt{{\left(p_{i+1,j}-p_{i,j}\right)}^{2}+{\left(p_{i,j+1}-p_{i,j}\right)}^{2}}$$

The gradient picture shows the change in pixels and helps the construction process start from positions that require more attention. It is obvious that a pixel in the gradient picture may also contain the information of the below pixels and those on its right; thus, it also offers a two-dimensional relationship between neighboring pixels, which greatly reduces misactivation. Figure 3 compares the blocks captured by the kernels with and without the gradient layer. 

The green blocks in the first column show the feature that a kernel represents. The mid-column in the table shows where the kernel will be activated if we add gradient information to the kernel; the right column shows where the kernels without gradient information will be activated. It could be seen that kernels without gradient information may be activated in more places and thus cause heavier computation. In addition, they are activated repeatedly in neighboring pixels, so they make little contribution to recognition.

#### 3.1.2. Kernel Maker

The nodes in the first layer are called kernels in this paper. They deal with pixel values and capture certain features. Each node has a unique ID, so it can easily be found when it is required by segments in higher layers. When a node is activated at different locations, it is treated as a different node and can easily be distinguished by its indexes. 

In kernels, the Pearson correlation coefficient reflects the distribution of a set of pixels, which probably (as the brain neuron system has not been well modeled at present) leads to feature recognition. Moreover, the value of the correlation coefficient is limited to [−1,1] (or [0,1] if we use its absolute value) regardless of the number of inputs, making it a good activation function for neural networks. It also does not require image normalization, as pixel distribution at the gray level can have little influence on the correlation coefficient. There is another benefit to this function. Since our purpose is to compare the similarity between the testing example and the learned pattern, it is convenient to obtain the filters’ parameters by directly copying them from the original pattern. Furthermore, the threshold could be simply set to 0.8, as this is a common mathematical standard for high correlations. In this way, we could avoid a gradient descent training process. Figure 4 shows how the kernels produced with this strategy capture features in the pictures.

The green blocks in the first column show the features the kernels represent and where they are copied. The second column shows the locations where the kernels are activated in the training picture and which nodes are learned by the latter layers. The third and fourth columns show the locations where the kernels are activated and the features that are captured by these kernels. It can be seen that the kernels constructed with this correlation coefficient-based strategy are able to capture the resembling features in testing pictures. The activated kernels in the second column, which is marked with purple blocks, will combine into segments in the latter layers.

The training example that is sent into the kernel maker contains three layers: the grayscale picture that offers the pixel values of the kernel; the gradient picture that not only offers pixel value but also tells the kernel maker where the most important information lies; and the efficient map that marks the area in which the kernels will be generated. Accordingly, the kernels have two layers that can deal with the grayscale and gradient pictures, respectively, making the actual size of the kernels 3 × 3 × 2. Kernels in this design are more like the multidimensional filters in traditional CNNs. Notice that, as a pixel in the gradient layer also contains the pixels below and on its right, only when these three pixels are all efficient is the gradient value of the pixel taken into the gradient layer of a kernel.

The kernel maker first divides the pixels into two categories: those with a gradient value and those without. Among those pixels with a gradient value, the maker randomly chooses coordinates as the centers of the kernels. Then, the maker builds 3 × 3 kernels around these centers. These kernels can be calculated to obtain their activated locations. The pixels covered by these activated kernels will be removed from the range for selection, so they will not be chosen repeatedly. After all the pixels with a gradient are covered, a pure kernel with all-zero weights is produced to capture the low-frequency information.

The pixels that are not efficient but are covered are marked as “i” in the kernel’s gradient layer. These “i” points will be ignored during calculation.

#### 3.1.3. Segment Maker

Segments deal with the locations of the activated nodes rather than their values. The wanted nodes are arranged in a certain order, normally from top to bottom and left to right. Segments can be of different sizes and shapes, similar to features captured by humans.

In segments, the correlation coefficient reflects the spatial relationships of child nodes. A segment consists of 2~6 child nodes (nodes that combine into a larger segment), which could either be kernels or segments. Its center is defined by Formula (2).
(2)$$\left\{\begin{array}{l} i_{center}=f_{int}(\frac{L_{i}(mid_{left})+L_{i}(mid_{right})}{2})\\ j_{center}=f_{int}(\frac{L_{j}(mid_{left})+L_{j}(mid_{right})}{2}) \end{array}\right.$$

$L_{i}$, $L_{j}$: The list that arranges the child node index in the order of i (vertical) and j (horizontal).

$mid_{left}$, $mid_{right}$: The index of a number in $L_{i}$ and $L_{j}$. This will be calculated by Formula (3). The indexes i, j start from 0.

$f_{int}()$: Obtains the integral part of the number.
(3)$$\left\{\begin{array}{l} mid_{left}=f_{int}(\frac{length_{c}-1}{2})\\ mid_{right}=f_{int}(\frac{length_{c}}{2}) \end{array}\right.$$

When a segment is built, the child nodes’ offsets from the center are recorded (Figure 5). We denote the number of child nodes as $length_{c}$.

For example, nodes 1–5 in Figure 5a are the child nodes that constitute a segment. We first calculate the center of the segment using the coordinates of the child nodes (higher row in Figure 5b). Then, the nodes are recorded with their offset from the center (lower row in Figure 5b). 

The segment maker continues to create new segments by combining the kernels/segments in former layers until all nodes converge into one node, just like the root of a tree graph. 

The steps to produce segments and layers are as follows:Group the kernels/segments that are connected to each other.For the first layer after the kernel layer, calculate the locations where the kernels are activated, then divide the area covered by the activated kernels into several connected sub-areas. The latter layers are built based on each sub-area.Randomly choose a start node; then, starting from this node, choose 1–5 nearby nodes to compose a segment. When choosing the nearby nodes, the segment maker first chooses one child node that is chosen by other segments in the same layer. After that, the maker chooses nearby nodes (≤4 nodes) that could be led to the start node by other connected nodes. The nodes that are not used by other segments will be given priority. As the status of nodes regarding whether they are chosen by other segments varies, the actual “steps” (number of nodes that lead the node to the start node) are not fixed. In most cases, we make sure the number of child nodes is between 4 and 6, but in cases where the number of nodes in the former layer is less than 4, the former nodes can also be formed into a segment. The pure kernel is not considered. Choosing a used node makes sure the segments in the new layer cover each other, so their locational relationship will be more reliable.In every sub-area, repeat the last step until there is only one node in each sub-area. For convenience, we denote the node that represents the pattern of a connected sub-area as the top node. Then, a final node is produced, following the same procedure.

The combination of kernels/segments can be seen in Figure 6.

The “weights” of kernels derive directly from pixels in the training pattern, and the weights of segments come from the locations of activated nodes in training images. The combination and layers are automatically decided. No further training process (especially no gradient descent procedure) is required to build a functional network.

#### 3.1.4. Network Architecture

Similar to existing neural networks, our network consists of several layers of kernels and segments. However, unlike those in CNNs, the segments in our model could have different sizes and shapes. The architecture is shown in Figure 7.

The input samples are scanned by the kernels in the first layer, and the locations of activated nodes are recorded in feature maps. The feature maps could also include the results coming from other layers if they are needed by the nodes in the next layer. The feature maps can be obtained, and then we can obtain a superposition map with all the activated nodes written in their activated location. This step is to simplify calculations and is not necessary if a better searching method is applied. The superposition map is then scanned by segments in the next layer.

In the final layer, the segments represent different structures of sub-networks and lead to end nodes. K final segments could be in N categories (N ≤ K), as there could be multiple structures in the same category. An end node is produced to represent the category of the learned example after the final node is built. A connection between the final node and the end node is established to identify the category of the learned pattern.

#### 3.1.5. Pre-Added Shapes and Shadow Nodes

This part focuses on future studies of multi-sample training. We added straight lines as basic components of the network. As the network is based on a combination of several features, these pre-added features, which are activated on the training sample, are taken as normal kernels/segments and used to build segments at higher layers.

In order to take more complicated functions into account, we designed shadow nodes to directly deal with logical operations. For example, “/”, ”|”, and ”\” will typically be treated as three different features, but when a shadow node defines these three features as the same shape, the network will identify them as the same type of segment.

### 3.2. Network Calculation

In this section, we show how the network built in the last section works and how the output is computed. Resembling the CNNs, kernels and segments slide throughout the picture to gather information in the window. The calculation of kernels is a bottom-up process, while the segments’ computing is more of a top-down process, as it involves deliberately searching and anchoring the needed child nodes.

The calculation process accompanies the correlation coefficient functions and the specially designed nodes. As shown in Figure 8, a gradient maker is also added after the input end of the network structure. The gradient maker generates a gradient picture according to the testing example and piles up the pictures. The two piled-up pictures are then sent to the network.

The pure kernel is calculated and recorded separately, as its feature map is only used in the check procedure, which will be described in Section 3.2.3. Checks and Restrictions. 

#### 3.2.1. Calculating the Neural Nodes

1.Calculating the Kernels

We use the Pearson correlation coefficient (shown in Formula (4)) as the synapse stimulation and neuron activation functions.
(4)$$R=\frac{\sum_{i=1}^{n}(P_{i}-\bar{P})(K_{i}-\bar{K})}{\sqrt{\sum_{i=1}^{n}{\left(P_{i}-\bar{P}\right)}^{2}\sum_{i=1}^{n}{\left(K_{i}-\bar{K}\right)}^{2}}}$$

The nine pixels in a kernel are put in sequence. The order is based on the pixel locations rather than the values to make sure the pixels at the corresponding location are paired up, as shown in Figure 9.

The numbers on the corresponding locations in the window and the kernel are paired, and the ineffective points in the kernels are skipped. The correlation coefficients of a grayscale picture and the gradient picture are calculated separately with the two kernel layers; the two results are then added with a weight of 0.45:0.55 (grayscale picture:gradient picture, as shown in Formula (5)). Only activated nodes (value ≥ 0.8) are recorded in a feature map, which will be used by higher layers. This enables the sparse activation of the nodes.
(5)$$C=0.45\times \left|C_{raw}\right|+0.55\times \left|C_{grad}\right|$$

2.Calculating the Segments

When the segment is produced, the pixels it covers are recorded according to the training sample and form a window that covers these locations. This window can be placed on the superposition map and used to collect the activated nodes needed by the segment. As the nodes in the former layer are sparsely activated, this collection process runs very fast.

The collected nodes can be filtered quickly to accelerate the calculation process. Although the segment shown in Figure 10 requires child nodes with four different IDs, it requires five nodes, as nodes in different locations are treated as different nodes. If we find one node 1, three node 2s, no node 3, and one node 4 in this sliding area, we are short two nodes. The maximum tolerance is calculated using Formula (6). If the collected nodes are less than the lowest requirement to activate a segment, the window will quickly move on to the next position. Apparently, in this position, the number of missing child nodes is too many to activate the segment, so the window quickly slides to the next position without the need for further calculation.
(6)$$tolerance_{missing}=f_{int}(length_{sequence}\times (1-threshold)+0.5)$$

threshold: The threshold for the correlation coefficient is set to 0.8, assuming each node makes an equal contribution to the activation.

$length_{sequence}$: The length of the list of child nodes needed to activate the segment.

If we find enough activated nodes with the IDs that are needed by the segment, we can move on to the calculation process. The nodes activated in former layers are recorded by their positions rather than values; thus, the traditional strategy in CNNs does not apply to our network. In order to properly deal with these indexes, we divide the information into two dimensions: distance and angle.

Firstly, we placed the found nodes in the same order as those recorded by the segment according to their IDs (assuming they are anchored, which we will describe in the following part). Then, we calculated the distances between nodes in the neighboring sequence (Figure 11) and recorded them as lists $L_{seg}^{d}$ (nodes recorded in the segment) and $L_{win}^{d}$ (nodes found in the sliding window). Their Pearson correlation coefficient was calculated by Formula (4), and the result is denoted as $C_{d}$.

The angle list records the angles of the half-lines between two nodes in the neighboring sequence turn (Figure 11 and Formula (7)). The angles are recorded in lists $L_{seg}^{\theta }$ (nodes recorded in the segment) and $L_{win}^{\theta }$ (nodes found in the sliding window). Their Pearson correlation coefficient is denoted as $C_{\theta }$.
(7)$$\Theta_{n}=F(\theta_{n}^{n+1}-\theta_{n-1}^{n})$$

$\theta_{n}^{n+1}$: The angle of the half-line connecting the No. n and No. n + 1 nodes in polar coordinates.

F(): Make sure the value is in the range of [−π, π]. For example, the $\Theta_{2}$ and $\Theta_{3}$ in Figure 11b are positive, while $\Theta_{1}$ is negative.

The nodes are placed in a list. The distances are calculated with two neighboring nodes (Figure 11a), and the angles are calculated with the half-lines between the neighboring nodes (Figure 11b). The value of this segment is calculated by Formula (8). Accordingly, the threshold should be squared by 0.8, as it is the product of two correlation coefficients (Formula (9)).
(8)$$C_{seg}=\left|C_{d}\right|\times C_{\theta }$$
(9)$$threshold_{2d}=threshold_{1d}^{2}$$

If more than one node with the same ID is needed or found, an anchoring process is applied to pair the nodes. This process is the most time-consuming aspect of the whole process, and we are still working on speeding it up.
(10)$${\vec{P}}_{pic}=\vec{M}{\vec{P}}_{seg}$$

The anchoring procedure is an affine transformation process that maps the segment into the testing block. As shown in Formula (10), the transform matrix $\vec{M}$ maps the index matrix ${\vec{P}}_{seg}$ into the expected index matrix ${\vec{P}}_{pic}$ in the picture. If only one node with a certain ID is needed by the segment and only one node with this ID is found in the window, these two nodes are considered “anchored”. If more than three pairs of anchored nodes are found, matrix $\vec{M}$ can be solved, and the unanchored nodes in the segment can be mapped into their expected locations in the picture. The nodes collected in the window with the same ID as those needed by the segment that are nearest to the expected index can be found. The paired nodes are anchored until all the needed nodes are anchored (Figure 12).

The segment first anchors three pairs of nodes and calculates the expected locations of the remaining nodes. The other nodes that are close to the locations are then anchored. The nodes found in the sliding window with the same IDs as those needed by the segment are shown in Figure 12b. Nodes 2,3,4 are single nodes in both the segment and the window, so they are directly anchored (Figure 12c); calculate the expected locations of Node 1 and anchor them (Figure 12d). Compute the segment and, if the segment is activated, break the traversal.

In most cases, however, more than one node with a certain ID is needed or found. This is when an ordered traversal is applied. The nodes will first be arranged in the order of the number of times nodes with the same ID were found in the window, as we humans would start from the most distinct feature. Then, the nodes needed in the segment are listed in the same order (Figure 13). After the two lists are made, the first two nodes can be anchored according to their IDs. Using these two pairs of nodes, we can calculate their rotation, shift, and scale to manually create a transform matrix. The expected locations of the remaining nodes are computed with the matrix. Then, the rest of the nodes are anchored according to the expected locations. When the nodes are anchored, their correlation coefficient can be calculated using function 4.

#### 3.2.2. Area Back-Tracing

There are three types of area back-tracing in our design: (1) minimize back-tracing, which is used for positioning and shape checks; (2) back-tracing with prediction, which is used for connectivity checks; (3) back-tracing for activated kernels and segments as templates, which is used to decide if enough elements are activated.

The anchored nodes that activate the segments are recorded in a map, so the tracing process resembles searching a bunch of leaves, starting from the same branch in a tree diagram, although some nodes might cross over each other or be re-used (Figure 14).

Minimize back-tracing: Only activated nodes are traced, ignoring the inactivated ones, which returns pixel-level information.With prediction: Activated nodes are traced directly; the positions of inactivated nodes are calculated according to the affine transformation mentioned above, returning pixel-level information.Template tracing: Back-tracing to the kernels that are activated is carried out to return the number of activated kernels that are required by the network.

#### 3.2.3. Checks and Restrictions

The checks presented below alleviate confusion between similar patterns. However, these checks slow down calculations as they repeatedly back-trace activated areas of nodes.

Rotation check: A rotation restriction is applied to the anchoring process to eliminate features in the wrong direction; detected features with a rotation larger than 60 degrees will be defined as inactivated. This is to ensure that the features are kept in a relatively stable direction. In this paper, we set the threshold of rotation limits to π/3, as the grid cells in the human brain form 60 degree connections.Shape check: Child nodes in the lower layers (at a certain check depth) of a detected feature (segment that passes the correlation calculation) will be traced, and their locations will be arranged in a line according to a segment calculation rule regarding similarity. These locations will be checked by function 8, and only nodes that pass the check will be defined as activated. The check depth is set to 2, as this meets our expectations for the experiments and keeps the resource cost reasonable (Figure 15).

Connectivity check: The child nodes of a detected feature will be back-traced to determine the area they actually cover. Their connectivity should remain the same as that registered in the segment in which they were built. Notice that child nodes that are connected by the same block in the pure layer can also be treated as connected.

## 4. Results

This model was initially designed for one-shot simple target detection in our project; therefore, we first tested it on the dataset we obtained in our project. As the work is based on a similarity-comparing idea, it should be able to match patches like those in the Brown–Yosemite dataset. We also tested the network on part of the Mnist dataset, even though the network was only designed to capture the learned pattern. As introduced in Section 2 on related works, most one-shot detecting networks rely on the pre-training of large datasets (or a mass of tasks), while ours is trained/constructed with only one training example, without any pre-training or gradient-based training, so it is hard to make a fair comparison. However, we listed the results of the Siamese Mast R-CNN and the method in [23]. It should be noted that, as the network’s construction relies on a random choice of start pixels, which results in fluctuations in performance, the tested networks are one possible establishment of each experiment.

The experiments and results for the dataset that we obtained in our project, Brown–Yosemite and part of Mnist, to prove its effectiveness are as follows:

### 4.1. Experiments on Datasets in Our Project

We tested our design on some videos and pictures we took in a real-life scene of our project. The main task of our project is to detect if the target is in the picture or video and, if yes, to output its location. All the networks are constructed with a single training sample without being pre-trained on a large dataset. Also, they are constructed without a hand-crafted dimension or layer number.

We set up targets in the field to check if the marked point is moved. As the target is set against a relatively stable background to mark a certain terrain and monitored by a fixed camera, every frame from the videos taken in our project looks alike. However, the leaves and water in the background could be confusing, and the illumination changes according to the time and weather.

#### 4.1.1. Test on Targets with a Simple Background

This video, with 643 frames, was taken in an experimental environment. The target was set in a room with a simple background and occupies a large proportion of the picture (Figure 16).

The efficient area occupies a relatively large portion of the picture and offers good features for training (Figure 16b). We took the first frame as the training picture (Figure 16a) and the rest of the frames as testing pictures. This setting was chosen to match the newly started projects that were tuned using earlier videos. The number of nodes and the network layers are automatically decided by the net constructor. As the features are chosen stochastically, the network and result may vary using the same dataset.

The output area of the detected object can be perceived, although the results are not state-of-the-art (SOTA) (Figure 17). We created a 2 × 2 down-sampling, and the area back-tracing was clearer with fewer misactivated kernels (Figure 17b). The pixel statistics are shown in Table 1.

Although the testing frames look very similar to the singular training frame, the pixel values change due to exposure and the slight movements of the target. As the network is limited, containing only the information from one training example, it may fail to capture features with more combinations. At the 3 × 3 scale, a few changes in pixels may cause drastic changes in the features, which will lead to detection failure in the kernels. This partially explains why the results obtained using the down-sampled pictures may be better, as this “smooths” pixel noise.

Furthermore, considering the construction of the kernels, the centers of the 3 × 3 kernels have to fall into the efficient area. Additionally, the non-‘i’ pixels have to be greater than three to enable coefficient calculation. Therefore, the area back-tracking may have a pixel loss problem following the construction procedure.

#### 4.1.2. Test on Targets in Real Scenery

In most cases, the target is standing outdoors with more complicated surroundings, and the target cannot occupy areas as large as those in experimental environments. In this project, we place a target against a background and among scenery with plenty of plants, and the target occupies a relatively small area (Figure 18). The network could resist most of the interfering context and return a rough area. Just like the first experiment, we took the first frame as the training sample and ran the network on the rest of the video.

The detected areas are shown in Figure 19, and the pixel statistics are shown in Table 2. Against a complicated background with leaves and trees, the network succeeds in capturing the small target and returns an approximate area. Although the IOU drops, it is easy to draw the contour of these pixels to mark the location of the target. As the network only relies on the use of one frame for training, it fulfills the need for a newly started object-detecting project that has unconventional targets and no training samples.

As the efficient area is smaller, with fewer activated kernels, the network scale is smaller than in the first experiments. Accordingly, the features that could help capture the pattern are limited. The decreased performance in Figure 19a may be caused by: (1) insufficient features to distinguish the target from the background; (2) stochastically chosen features and combinations; (3) a resolution that could neither “smooth” pixel noise nor provide evident features; or (4) complex conditions compared to an experimental environment.

We used the segmentation results of the Siamese Mask R-CNN and the network in [23] for comparison (Figure 20).

Both networks achieved better performance with color pictures and failed when grayscale pictures were used. As our design in this version works only on grayscale pictures, we ran these two networks on both grayscale (upper row in Figure 20) and color pictures (lower row in Figure 20) to obtain a better comparison.

#### 4.1.3. Test on Targets against Novel Scenery

In the former tests, the training examples and the testing pictures were from the same video with a similar background. In order to test the model’s ability to capture a similar object when placed against different backgrounds, we ran the same network that was used for the last video with different scenery and found that it was capable of locating a similar target against different backgrounds without pre-training or other adjustments (Figure 21). The pixel statistics are shown in Table 3.

A target of a different size and against a different background could also be detected and located by the same network that was constructed in the former experiment, showing that the network has a degree of transferability. Although it is limited by the information offered by the single training sample, this network shows a certain capacity for generalization.

The current back-tracing program does not include pure colored blocks in a pure layer that connects neighboring features (although it passes the connectivity check with these blocks), producing gaps between features when the network is used in a larger picture. However, a contour could be drawn to show the location of the target.

We also tested this picture on a Siamese Mask R-CNN and the network in [23], using both grayscale and color pictures (Figure 22).

As our dataset for this project is limited, we tested the network on pictures of sparrows to show its ability to segment multiple objects in diverse locations. The network is shown in Figure 23a, with an efficient map shown in Figure 23b. The learned pattern that is marked by the efficient map is a front-sided sparrow. The network is then tested, as shown in Figure 23c. Figure 23d shows that the area of the two front-sided sparrows is roughly marked, while half of the side-view sparrow is missed.

### 4.2. Experiments on the Brown–Yosemite Dataset

The pictures in the Brown dataset are 64 × 64 grayscale pictures constructed by back-projecting 3D points to the pictures taken during photo tourism. In Yosemite’s patch list, there are 500 pairs of matching pictures and 500 pairs of non-matching pictures.

We set the efficient area as the full picture, constructed the network on the first picture of a pair of patches, and tested it on the second one. For each pair of patches, we repeated this procedure. The network was then discarded. Therefore, the networks are limited to a pair of pictures and do not take advantage of other pictures in the same domain. Earlier, these networks were referred to as disposable networks.

We obtained 928 results, including 461 non-matching patches and 467 matching patches (Table 4).

Although the results are not state-of-the-art, it should be noted that this network only works on one-shot samples without any pre-training.

### 4.3. Experiment on Part of the Mnist Dataset

The Mnist dataset is a large dataset that consists of 70,000 handwritten numbers, among which there are 60,000 training samples and 10,000 testing samples. It is widely used as a standard and simple dataset for classification.

Although we tested the network using the Mnist dataset, our network was not initially designed for classification. Also, the network only recognizes certain combinations of learned objects and roughly divides the samples into positive and negative categories. It can be foreseen that if a learned pattern is contained in the testing picture, then this may lead to an incorrect classification, such as “1” in “7”. With more training pictures (which could be gathered during the project, as long as there are fewer than 10) and a proper training process (which will be described in the next paper), this network will be more stable and able to fulfill classification tasks (as will be accomplished in future studies).

We created efficient masks with pixels that have values larger than 0.03 for the training samples. Then, we built nine parallel networks with nine samples from different categories. The line number of the training samples is listed in Table 5, as the samples are closer to the standard numbers.

**Table 5 sensors-23-09188-t005:** Line numbers of the training samples from different categories. Category 2 is not recorded, as there are two different ways to write a “2” that cannot be learned using a single sample (Figure 24).

Cates	0	1	2	3	4	5	6	7	8	9
Line Number	211	9	--	434	93	48	389	104	709	134

We tested the network on samples 50,000–51,000; the results are shown in Table 6.

The ground truth is slightly stronger than the picture looks, as it strengthens some darker pixels, which are also included in efficient areas. As shown in Figure 25, both samples are successfully recognized as “3”. This shows that our design could deal with a certain degree of shapeshifting.

Table 7 shows a comparison between existing designs and our design.

The correlation coefficient-based strategy enables our design to obtain parameters without the gradient descent back-propagation process, which enables the network to quickly “learn” the features of a simple pattern without pre-training. Compared to the existing OSD methods mentioned in Section 2 on related works, our design is cheaper to train as it does not require the collection of training pictures and intense pre-training computation using mass data, and thus is more friendly to newly started projects with few training pictures. However, as all the information needed for detection is provided by a single training example, the network greatly relies on this one training sample, which results in limited performance. As no knowledge is accumulated in the training process and only one example is introduced in this paper, the design is temporarily limited in construction and test procedures.

Although this network achieved one-shot detection without pre-training and the gradient-based learning process, this achievement is not mature and has not been well studied compared to existing CNNs. There are still problems that degrade the performance: (1) the information that can be obtained from a single training picture is limited, so the learned features may not be adequate to deal with the test examples if the target is too different from the learned pattern. (2) Pixel-level noises may cause drastic changes at the 3 × 3 scale, causing the kernels to fail to capture the features. (3) Although we applied overlapping kernels/segments to strengthen stability, the performance of networks built on the same training example may vary because of the stochastic selection of features and the combinations of network construction procedures, which may introduce randomness to the performance. Luckily, as the construction process of this design is cheap, we could easily filter better structures with validation examples. However, this issue should be fixed in future studies. (4) In the construction/training phase, the centers of the kernels have to fall within an efficient area, marked by the efficient map, while the kernels with fewer than three non-‘i’ pixels are deleted to avoid the inefficient calculation of correlation coefficients, causing pixel loss in the covered area. (5) The pure layer is not used in area back-tracking, leaving gaps between the detected features. (6) As we did not apply GPUs or multiple threads to our program, the calculation could be rather slow. This could have severe consequences when the efficient area is large, as the network scale is likely to be large and contain more nodes. We leave these problems to future work.

## 5. Conclusions

In this paper, we propose a correlation coefficient-based neural network that performs one-shot detection and positioning on simple targets. Compared to existing methods, our model requires no pre-training thanks to the application of the Pearson correlation coefficient. As no pre-training of mass data is applied, our model has a cheaper construction process and is not limited by domain gaps and a fixed backbone. Also, the model is designed to require no handcrafted parameters, such as dimension and layer numbers, and to organize its structure according to the complexity of the learned sample. Furthermore, our design has better interpretability compared to existing models for the easy back-tracing of activated nodes. As far as we know, this is a new attempt to apply a non-training parameter-gaining strategy that directly copies from the training samples in neutral networks and uses a correlation coefficient as the activation function. Experiments on the dataset in our project, the Brown–Yosemite dataset and part of the Mnist dataset, show that it fulfills the goal of quickly starting a simple target-detecting project that lacks training data and outputs the targets’ locations.

At present, this design is still limited by the use of a single training picture and a random choice of kernels. In future studies, a structure-based training strategy that uses more training samples will be implemented to augment the inter-class and between-class comparisons. This training strategy enables the network to fulfill more complicated tasks. More training pictures also enable knowledge accumulation, which helps to recognize targets with more transformations; therefore, the network will be more stable and achieve better performance. Differing from most existing neural networks, which are trained by changing their parameters in filters, our network is trained by changing the structural connections. We hope that this design can offer a new perspective on machine learning.

Another problem to be fixed in future work is how to deal with the background. In this version, the background is simply ignored. In future work, the background will be taken into consideration to allow for more details and greater distinction. This will alleviate the “1 detected in 7” problem.

## Figures and Tables

**Figure 1 sensors-23-09188-f001:**
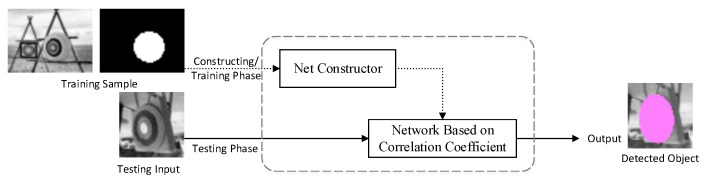
Architecture of the correlation coefficient-based model.

**Figure 2 sensors-23-09188-f002:**
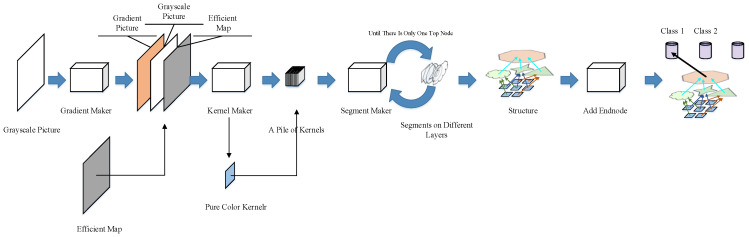
Structure of the net constructor.

**Figure 3 sensors-23-09188-f003:**
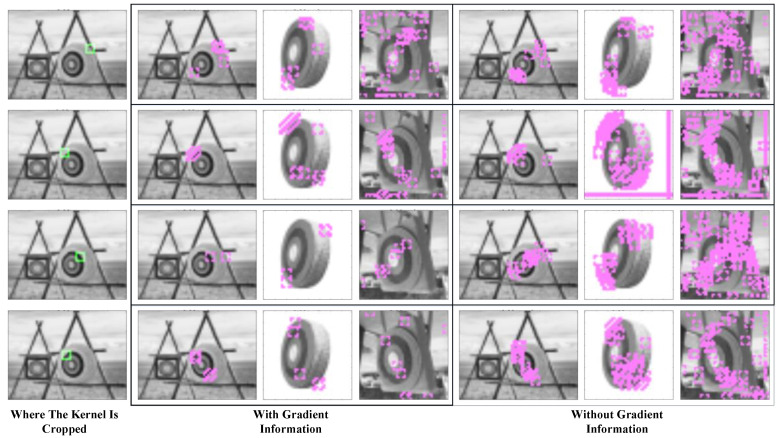
Comparison of kernels with and without the gradient layer. The green blocks show the locations where the features are picked. The purple blocks show where the features are activated.

**Figure 4 sensors-23-09188-f004:**
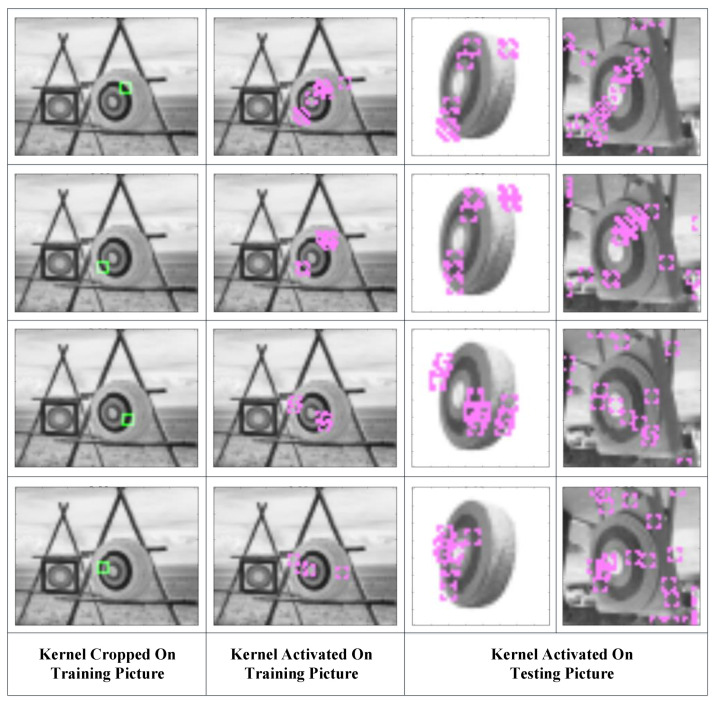
Kernels that were constructed (green blocks in the first column), learned (purple blocks in the second column), and activated in the testing sample (purple blocks in the third and fourth columns).

**Figure 5 sensors-23-09188-f005:**
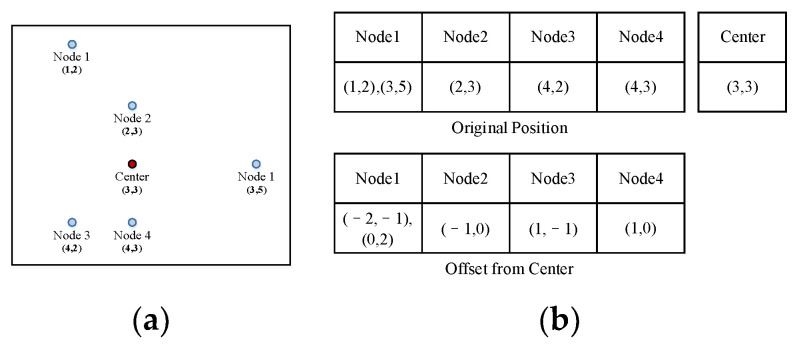
Record of segments by offsets. (**a**): Child nodes of a segment; (**b**): The original and recorded positions of the child nodes.

**Figure 6 sensors-23-09188-f006:**
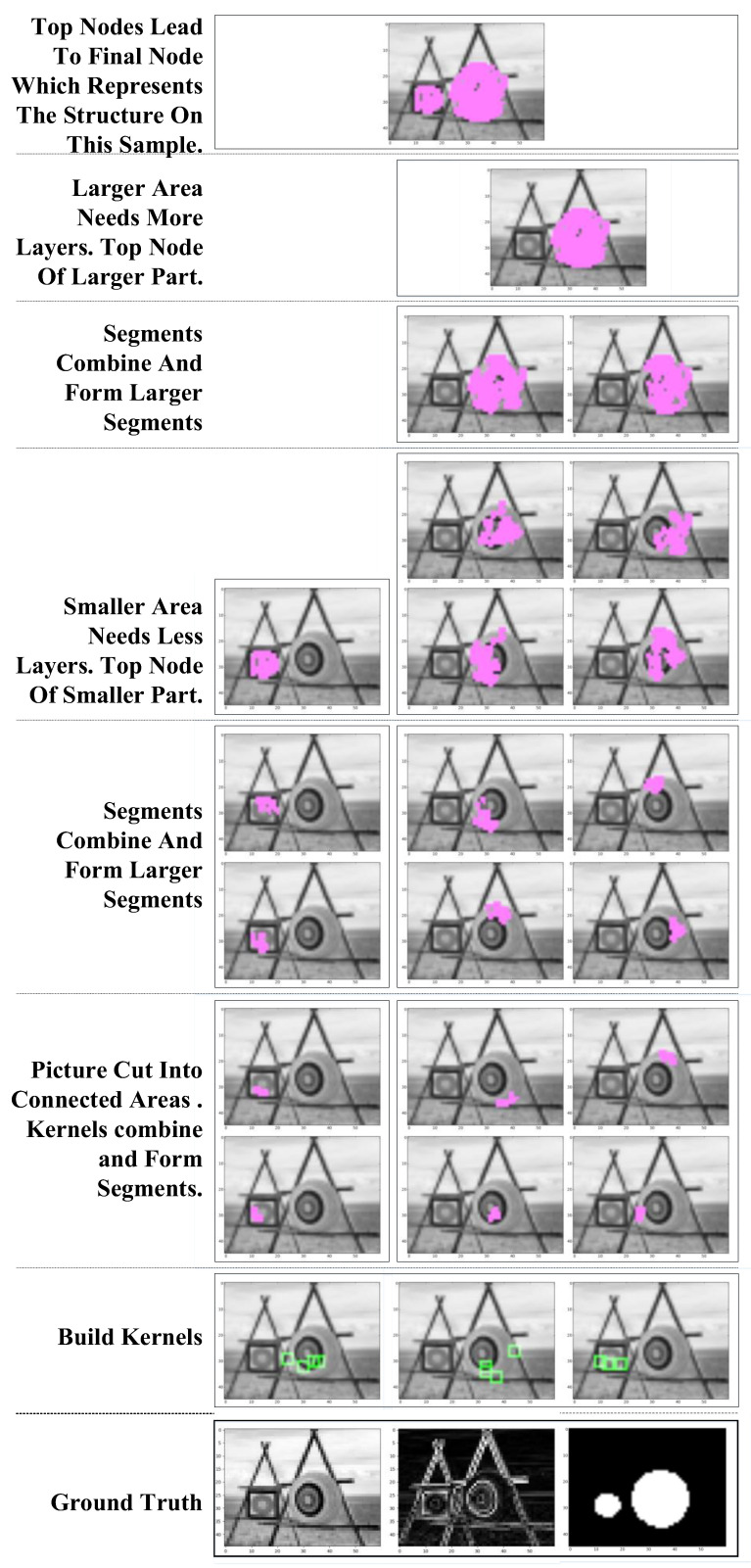
Kernels/segments combined into segments in higher layers. Green blocks show where the kernels are picked. Purple areas mark the original locations and shapes of segments.

**Figure 7 sensors-23-09188-f007:**
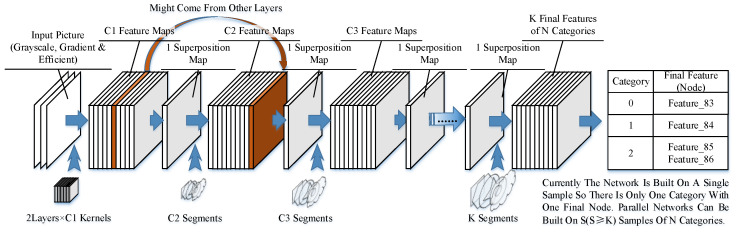
Network structure.

**Figure 8 sensors-23-09188-f008:**
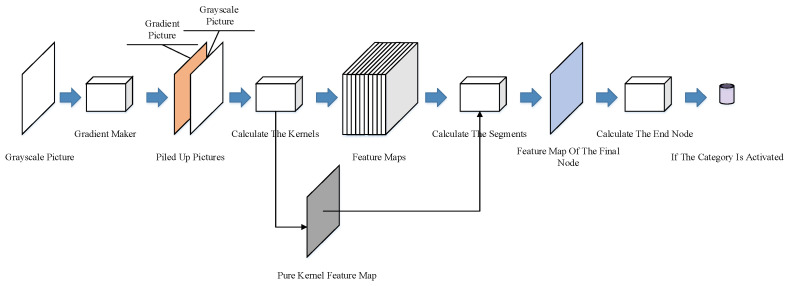
Network calculation process in the testing phase.

**Figure 9 sensors-23-09188-f009:**
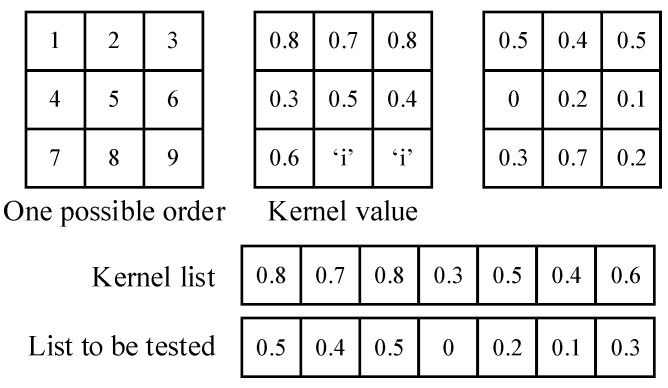
Calculation of kernels. Pixels not in efficient map are marked as “i”.

**Figure 10 sensors-23-09188-f010:**
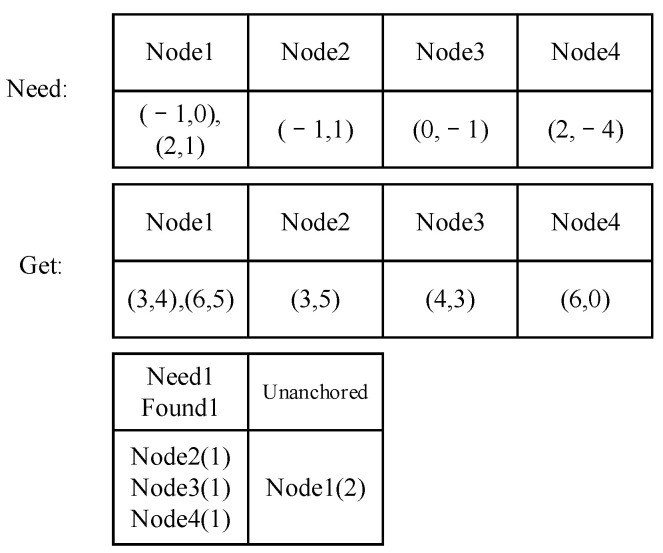
Filtering the nodes needed.

**Figure 11 sensors-23-09188-f011:**
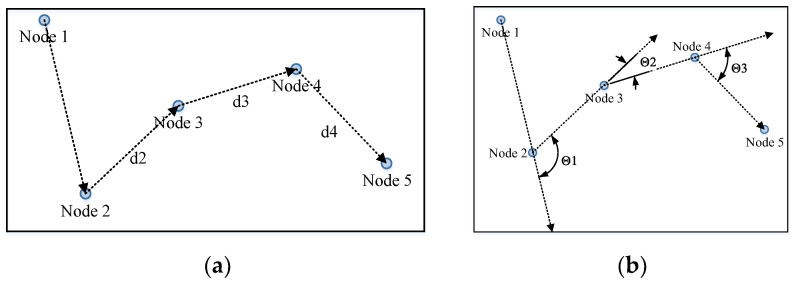
Calculation of segments. (**a**) Calculation of distance between two neighboring nodes; (**b**) calculation of angles between two neighboring nodes.

**Figure 12 sensors-23-09188-f012:**
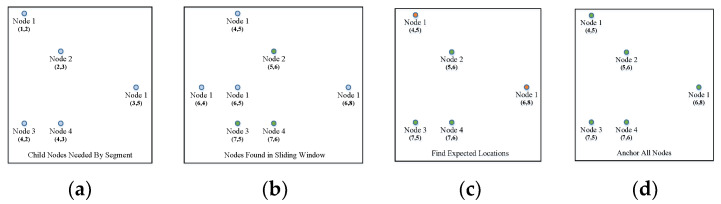
Anchoring procedure in the segment calculation. (**a**) Nodes needed by the segment; (**b**) nodes found in the sliding window with the same ID as the needed nodes; (**c**) anchoring nodes 2,3,4; (**d**) other anchoring nodes.

**Figure 13 sensors-23-09188-f013:**
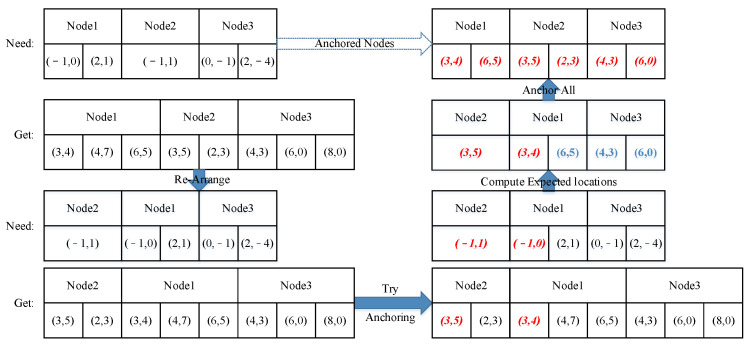
Anchoring process when there are more nodes than needed. Anchored nodes are marked in red color. Expected coordinates of unanchored nodes are marked in blue.

**Figure 14 sensors-23-09188-f014:**
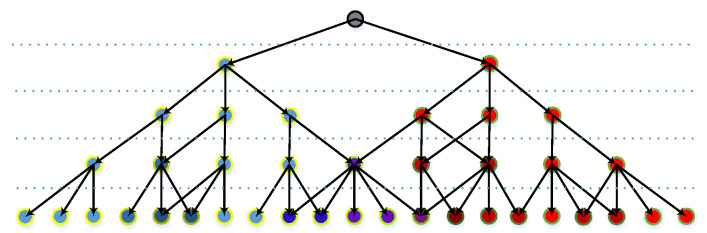
Tree-like diagram recording nodes that activate a feature. Different color shows the nodes are recorded in different branches. The color varies if a node is recorded in multiple branches.

**Figure 15 sensors-23-09188-f015:**
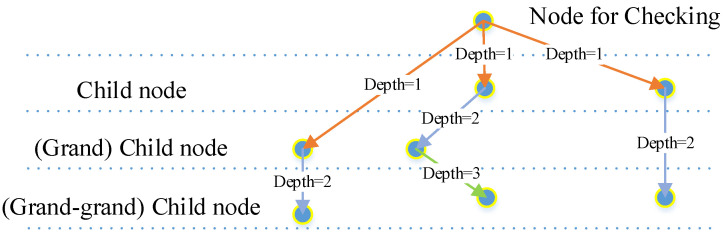
Check depth of child nodes: as the child nodes could come from different layers, the check depth is decided by their real level, counting backwards from the node that is to be checked. Orange arrows: depth=1; blue arrows: depth=2; green arrow: depth=3.

**Figure 16 sensors-23-09188-f016:**
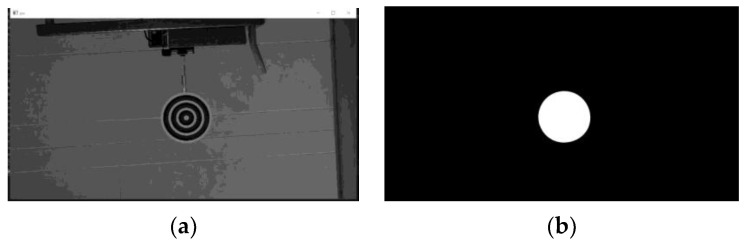
Training frame. The rest of the frames are from the same video in the same scenario: (**a**) grayscale picture; (**b**) efficient map.

**Figure 17 sensors-23-09188-f017:**
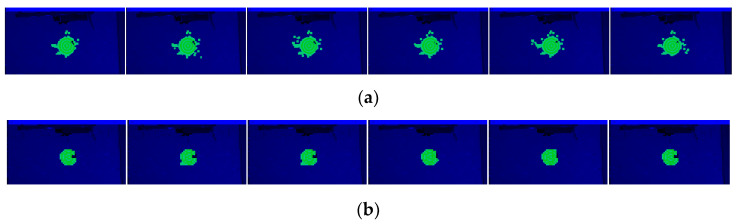
(**a**) Six results from the 643-frame video; (**b**) six results from the same video with 2 × 2 down-sampling.

**Figure 18 sensors-23-09188-f018:**
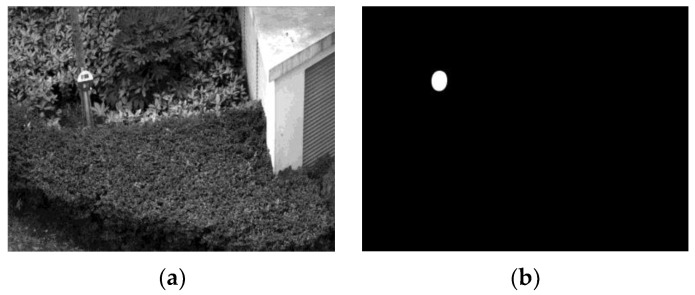
(**a**) Training frame; (**b**) a relatively small efficient area, which could be as small as 75 effective pixels with 2 × 2 downsampling.

**Figure 19 sensors-23-09188-f019:**
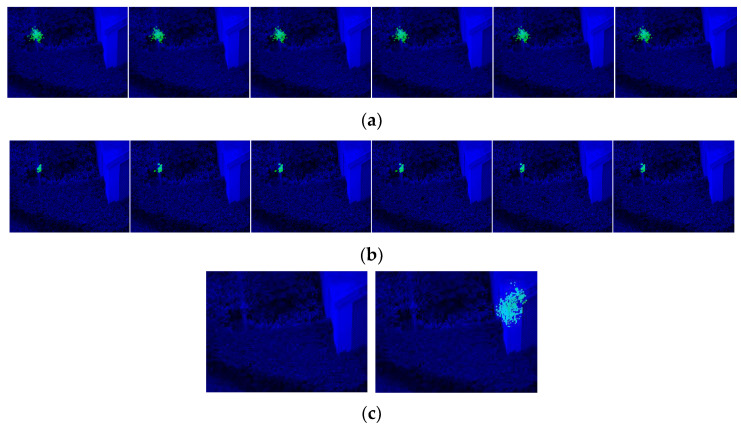
(**a**) Results for 240 × 320 resolution pictures. (**b**) Results for 2 × 2 downsampling pictures with 75 effective target pixels. (**c**) Results for unsuccessful structures with detection failures.

**Figure 20 sensors-23-09188-f020:**
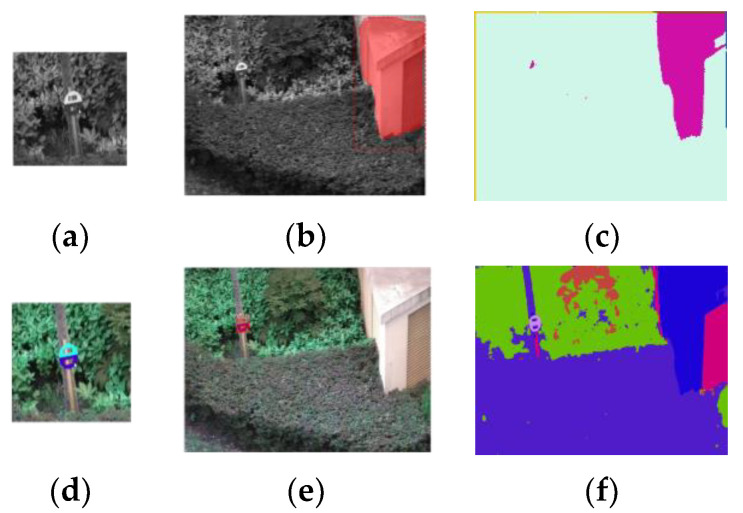
Segmentation results of a Siamese Mask R-CNN and the network in [23]. (**a**,**d**) Reference for the Siamese Mask R-CNN. (**b**,**e**) Segmentation results of the Siamese Mask R-CNN, segmentation shown in red. (**c**,**f**) Segmentation results of the network in [23]. The upper row shows the results for the grayscale pictures, and the lower row shows the results for the color pictures.

**Figure 21 sensors-23-09188-f021:**
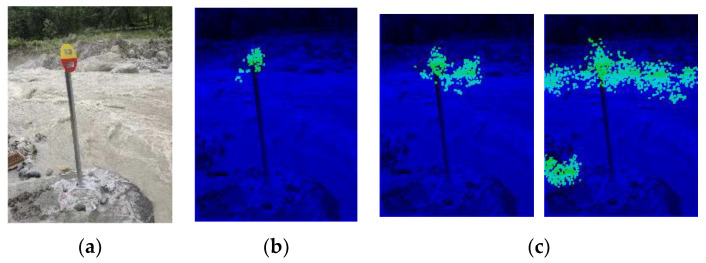
(**a**) A 244 × 185-sized picture with a similar target and a different but complex background. (**b**) Positioning results. (**c**) Results of unsuccessful structures with detection failures.

**Figure 22 sensors-23-09188-f022:**
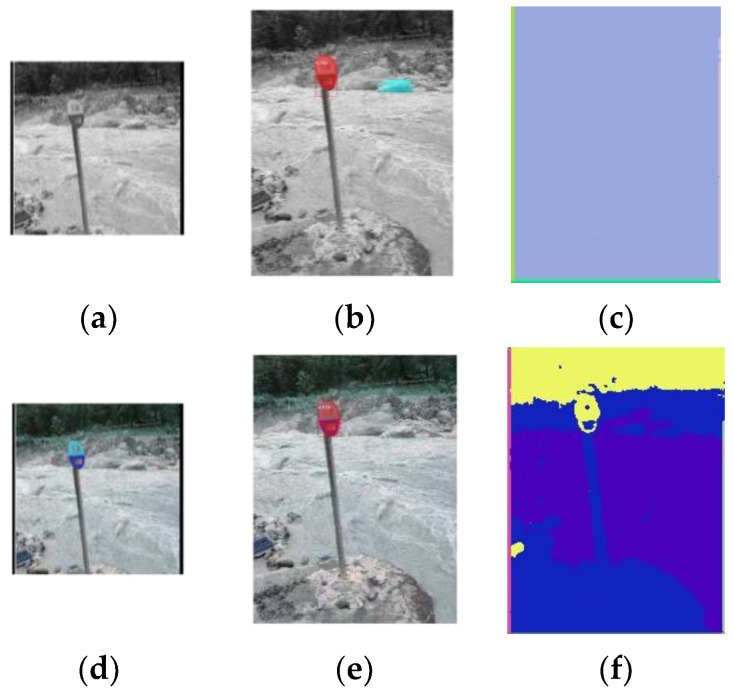
Segmentation results of the Siamese Mask R-CNN and the network in [23]. (**a**,**d**) Reference for the Siamese Mask R-CNN. (**b**,**e**) Segmentation results of the Siamese Mask R-CNN; the segmentation is shown in red. (**c**,**f**) Segmentation results of the network in [23]. The upper row shows the results for the grayscale pictures, and the lower row shows the results for the color pictures.

**Figure 23 sensors-23-09188-f023:**
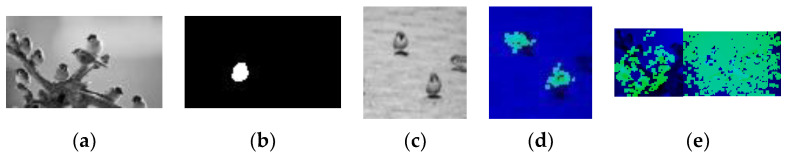
Results obtained against a novel background. (**a**) Training example; (**b**) efficient map; (**c**) testing picture; (**d**) detection results. (**e**) Results of unsuccessful structures with detection failures.

**Figure 24 sensors-23-09188-f024:**
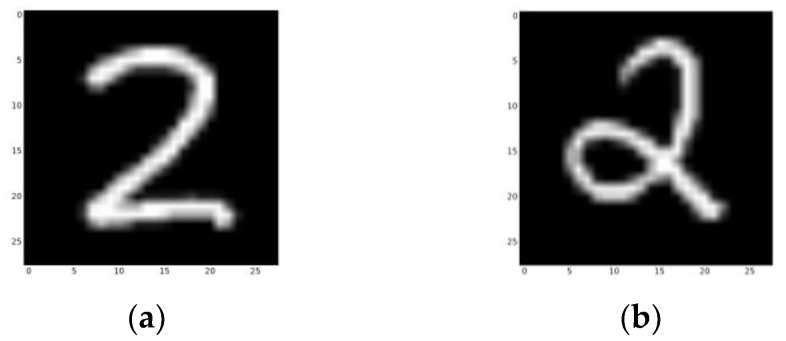
(**a**) The 1439th line in the Mnist dataset. (**b**) The 1897th line in the Mnist dataset. Both are examples of category 2.

**Figure 25 sensors-23-09188-f025:**
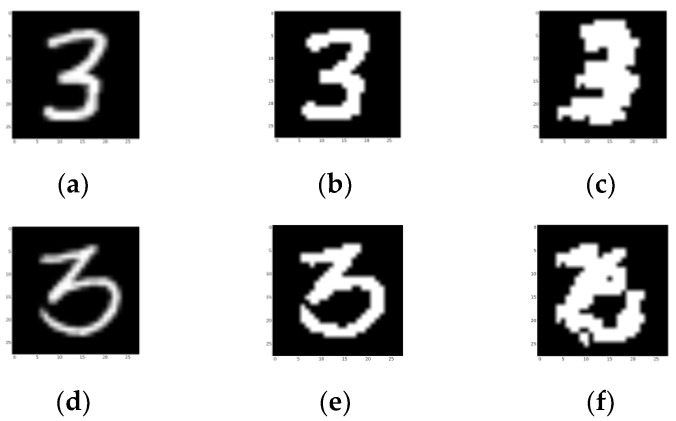
Segmentation results of a successful detection in category 3. (**a**,**d**) Testing picture. (**b**,**e**) Ground truth. The final nodes are activated in three locations. (**c**,**f**) Segmentation results of the network. (**d**–**f**) Segmentation results of the three final activated nodes.

**Table 1 sensors-23-09188-t001:** Average area detection results of the six frames shown in Figure 17. TN: true negative; FP: false positive; FN: false negative; TP: true positive; IOU: intersection over union. As there are no bounding boxes in our design, the area is calculated by the pixels that are covered.

	Ground Truth		Detected		Expect False		Expect True		Accuracy	IOU
	Positive	Negative	Positive	Negative	TN	FP	FN	TP	__	__
**(a)**	479	13,601	633	13,447	13,419	182	29	451	98.50%	68.21%
**(b)**	107	3413	97	3424	3411	2	12	95	99.60%	87.04%

**Table 2 sensors-23-09188-t002:** The average area detection results of the six frames shown in Figure 19.

	Ground Truth		Detected		Expect False		Expect True		Accuracy	IOU
	Positive	Negative	Positive	Negative	TN	FP	FN	TP	__	__
**(a)**	304	76,496	658	76,142	76,103	393	39	265	99.44%	38.00%
**(b)**	75	19,125	53	19,147	19,116	9	31	44	99.79%	52.84%

**Table 3 sensors-23-09188-t003:** The result of area detection against a novel background.

Ground Truth		Detected		Expect False		Expect True		Accuracy	IOU
Positive	Negative	Positive	Negative	TN	FP	FN	TP	__	__
548	44,592	385	44,755	44,510	82	245	303	99.28%	48.10%

**Table 4 sensors-23-09188-t004:** Results of the Brown–Yosemite dataset.

	Matching Patches	Non-Matching Patches
Number of all the samples	467	461
Number of samples detected match	True Positive: 380	False Positive: 96
Number of samples detected non-match	False Negative: 87	True Negative: 365
Recall	81.37%	79.18%
	False Negative Rate: 18.23%	False Positive Rate: 20.82%
Accuracy	80.28%	

**Table 6 sensors-23-09188-t006:** Results from the Mnist dataset. Category 1 has a high false positive rate because its pattern is contained in many other numbers, such as 2, 7, and 9.

Category	Positive Examples	Negative Examples	TN	FP	FN	TP	Accuracy
0	100	900	827	73	39	61	88.8%
1	107	893	514	379	15	92	60.6%
3	105	895	852	43	67	38	89.0%
4	89	911	698	213	27	62	76.0%
5	91	909	700	209	58	33	73.3%
6	104	896	849	47	64	40	88.9%
7	105	895	757	138	25	80	83.7%
8	108	892	872	20	86	22	89.4%
9	98	902	791	111	68	30	82.1%

**Table 7 sensors-23-09188-t007:** Comparison of different methods.

	Siamese Networks	Transfer Learning	Meta-Learning	Our Method
One-shot without pre-training	√	×	×	√
Positioning	√	√	√	√
Auto-organized structure for different complexity	×	×	√	√

√ shows the design conforms to this property and × shows the opposite.

## Data Availability

Data are contained within the article.

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
