# Peer review of "One-Shot Simple Pattern Detection without Pre-Training and Gradient-Based Strategy"

_sensors, 2023, doi:10.3390/s23229188_

Round 1
Reviewer 1 Report
Comments and Suggestions for Authors
Title should be more precise; it should be made clearer that the approach is limited to simple shapes and patterns.
The method is laid out pretty clearly, but there are a lot of repetitions on some statements, e.g. that you train the network with a single sample.
In my opinion, the experiments you conducted are not appropriate and do not show the limitations of your approach. In 4.1.1, you test the algorithm on 643 frames without giving information about what the original video shows. In the 6 example frames shown to the reader, every frame looks the same.
You explain repeatedly that it is hard to compare your work to others, but you should show the performance of other algorithms one your dataset to give more perspective about the raw resulting numbers.
The dataset should be more diverse and include different scenes, maybe changed pattern size and position.
Experiment 4.1.3 should be conducted in greater detail, as the performance of detecting targets in novel backgrounds is important.
The following two experiments (4.2 and 4.3) are not relevant to your algorithm and cannot show the localization part but only the detection.
You list performance problems in your conclusions, but the paper does not include any information about the calculation performance of your network.
As your algorithm is tailored to recognize and locate patterns, you should cite more literature about classical approaches to pattern recognition.
Comments on the Quality of English Language
Overall, English must be drastically improved.
There are a lot of typos and grammatically the comlete text must be revised.Author Response
Dear Reviewer:
Thank you very much for your constructive comments and valuable suggestions. We have taken all the comments into account, and have made major corrections in this manuscript. In the revised manuscript, the removed parts from the original paper are shown in revision mode, the added parts are highlighted in yellow, minor edition are marked in blue. All the comments are shown in the annotation mode. The point-by-point response is attached below, and the revised manuscript is enclosed, please see the attachment.
We really appreciate your concern of our manuscript and thank you very much for your comments.
Yours Sincerely,
Jun Su
J_su
Point-by-pont response:
- Title should be more precise; it should be made clearer that the approach is limited to simple shapes and patterns.
The title has been changed into “One-shot Simple Pattern Detection Without Pre-training and Gradient Based Strategy”.
- The method is laid out pretty clearly, but there are a lot of repetitions on some statements, e.g. that you train the network with a single sample.
Some minor edition has been made to alleviate this problem. The added parts are highlighted in yellow, removed part are marked in revision mode and minor adjustments are marked in blue.
- In my opinion, the experiments you conducted are not appropriate and do not show the limitations of your approach. In 4.1.1, you test the algorithm on 643 frames without giving information about what the original video shows. In the 6 example frames shown to the reader, every frame looks the same.
The frames look very similar indeed because the camera was monitoring a fixed target in a certain background in case it moves or disappears. Actually, the network in this paper could only handle simple situations. It only gets limited information and uses a very immature way to set up and calculate. We are still working on optimizing this method so it would be able to handle more complicated situations.
- You explain repeatedly that it is hard to compare your work to others, but you should show the performance of other algorithms one your dataset to give more perspective about the raw resulting numbers.
The segmentation results of Siamese Mask R-CNN and an unsupervised image segmentation network have been added. The former one is given a single reference picture in testing phase and outputs the detection result, which is resembling to our settings. It is pre-trained on mass data without mask annotations. The latter one uses no references and simply segment the elements of a picture.
- The dataset should be more diverse and include different scenes, maybe changed pattern size and position.
The performance is limited in real-scene pictures, but the experiments on Mnist dataset shows the network could tolerate a certain degree of shapeshift. It’s really a very good point that the position change should be shown in the experiments. Unluckily in the very limited data we got in our project the target seems to at similar position. So we tested it with a picture with sparrows.
- Experiment 4.1.3 should be conducted in greater detail, as the performance of detecting targets in novel backgrounds is important.
As our dataset from the project is so limited, we added a detecting result of sparrows in experiment 4.1.3.
- The following two experiments (4.2 and 4.3) are not relevant to your algorithm and cannot show the localization part but only the detection.
The segmentation results of Mnist dataset, both correctly and wrongly detected, are added in experiment 4.3. The experiment 4.2 is an appendant that enlarges the efficient area to full-sized pictures. We have added segmentation results of Mnist dataset to experiment 4.3.
- You list performance problems in your conclusions, but the paper does not include any information about the calculation performance of your network.
Results of detection failures are added after section 4.3.
- As your algorithm is tailored to recognize and locate patterns, you should cite more literature about classical approaches to pattern recognition.
YOLO series, Faster R-CNN, Mask R-CNN, Siamese Mask R-CNN and an unsupervised image segmentation network are added to citations.
Reviewer 2 Report
Comments and Suggestions for Authors
This paper the One-Shot target detection using gradient descent and pre-training. The topic of this research is interesting. However, the paper is not well structured and written. It is unclear what the research questions and research objectives, are and what the research gaps in the literature. The study applies many one-shot target detection using gradient descent and pre-training analysis methods. However, it is not clear why the authors conducted so many analyses. Are there any previous one-shot studies using gradient descent and pre-training? What are the findings consistent with previous studies? What new empirical knowledge added to the literature? Other minor comments are given below.
1. The introduction and related work explain the logical confusion and do not explain the problems to be solved in this paper. How do the descriptions in this paper relate to the research objectives? What are the limitations of traditional algorithms and other methods? Is it only a general discussion to solve the problem of target detection or segmentation? It is necessary to simplify and remove the concepts unrelated to the research in this paper. The problem should drive it. Starting from the problem, combined with the advantages of one-shot to solve the problems in the paper, it is recommended to further sort out.
2. In the training set, each category has samples, but only a small number of samples ( only one or several ). In this case, we can learn a generalized mapping on a larger data set or use methods such as knowledge graph and domain knowledge, then update the mapping on a small data set. The training set described by the author only uses one category, and the validation set also uses one category. The data set is currently small, and whether the small data set can support the author to train a robust network, and if there is a new category to join, it is necessary to retrain the network.
3. For the same attribute, in different categories, the performance of visual features may be very large. For example, zebras and pigs have tails, so in the category semantic representation, ' have tails ' are non-zero values, but the visual features of the two tails are very different. If the zebra is a training set and the pig is a test set, it is difficult to classify the pig correctly by using the model trained by the zebra. It is suggested that the author further supplements how to solve the problem of pivot point, semantic interval, and domain drift.
4. It is recommended that the author increase the omniglot dataset to verify the generalization of the proposed algorithm.
5. It is suggested that the author increase the number of training iterations and the accuracy effect diagram.
6. It is recommended to cite important literature in the past three years, which are used in deep learning small target detection. such as:
--Hybrid Behrens-Fisher- and Gray Contrast–Based Feature Point Selection for Building Detection from Satellite Images
--Constraint-Based Evaluation of Map Images Generalized by Deep Learning
7. How to solve the problem of low accuracy of segmentation boundary.
8. It is recommended that the author further increase the efficiency experiments with other methods to verify the efficiency of the proposed algorithm.
Comments on the Quality of English LanguageMinor editing of English language required.
Author Response
Dear Reviewer:
Thank you very much for your constructive comments and valuable suggestions. We have taken all the comments into account, and have made major corrections in this manuscript. In the revised manuscript, the removed parts from the original paper are shown in revision mode, the added parts are highlighted in yellow, minor edition are marked in blue. All the comments are shown in the annotation mode. The point-by-point response is attached below, and the revised manuscript is enclosed, please see the attachment.
We really appreciate your concern of our manuscript and thank you very much for your comments.
Yours Sincerely,
Jun Su
J_su@mail.sim.ac.cn
Point-by-point response:
This paper the One-Shot target detection using gradient descent and pre-training. The topic of this research is interesting. However, the paper is not well structured and written. It is unclear what the research questions and research objectives, are and what the research gaps in the literature. The study applies many one-shot target detection using gradient descent and pre-training analysis methods. However, it is not clear why the authors conducted so many analyses. Are there any previous one-shot studies using gradient descent and pre-training? What are the findings consistent with previous studies? What new empirical knowledge added to the literature? Other minor comments are given below.
Thank you very much for pointing out this fatal problem. We’ve reedited the Introduction and Related Works part. The edited parts are shown in revision mode in Word.
Our main attempt is to cut off the pre-training process that is applied in most one-shot detection networks. To achieve this, we come up with a correlation coefficient based network. Due to this design, the training/building process no longer relies on gradient-based training methods which have been applied on most neural networks. It may be the first time to use such a correlation coefficient based non-training strategy to get the parameters of a network, as far as we know.
- The introduction and related work explain the logical confusion and do not explain the problems to be solved in this paper. How do the descriptions in this paper relate to the research objectives? What are the limitations of traditional algorithms and other methods? Is it only a general discussion to solve the problem of target detection or segmentation? It is necessary to simplify and remove the concepts unrelated to the research in this paper. The problem should drive it. Starting from the problem, combined with the advantages of one-shot to solve the problems in the paper, it is recommended to further sort out.
The main purpose of this design is to cut off the gradient based training process and the pre-training procedure of most OSD networks. The edited parts are marked in revision mode.
- In the training set, each category has samples, but only a small number of samples ( only one or several ). In this case, we can learn a generalized mapping on a larger data set or use methods such as knowledge graph and domain knowledge, then update the mapping on a small data set. The training set described by the author only uses one category, and the validation set also uses one category. The data set is currently small, and whether the small data set can support the author to train a robust network, and if there is a new category to join, it is necessary to retrain the network.
In most widely used one-shot networks, training on a larger dataset and fine-tune it on smaller query set is an essential procedure. Actually, the main purpose of this paper is to cut down the training process on larger dataset and let the network learn from a single picture (or a few pictures in future version) in novel domains. That’s why we applied correlation coefficient rather than traditional convolution methods to deal with the pattern pixels and shapes directly. It is really challenging to train the network with so limited information, and it turns out that the network is not very robust, that’s why we had to filter better structures with validation pictures. In current version, when a new category joins, we just make a parallel network (a new one that works together with the old one) as no further training strategy is applied. In further study, a structure-based training method will be applied so the network could learn more information from more samples.
- For the same attribute, in different categories, the performance of visual features may be very large. For example, zebras and pigs have tails, so in the category semantic representation, ' have tails ' are non-zero values, but the visual features of the two tails are very different. If the zebra is a training set and the pig is a test set, it is difficult to classify the pig correctly by using the model trained by the zebra. It is suggested that the author further supplements how to solve the problem of pivot point, semantic interval, and domain drift.
It's indeed a very important and promising direction to be stressed. In designing phase, we did plan to add “joints” to the connected features. However, with a single picture, the network can hardly decide which part could be treated as rotatable or movable. For example, letter “J” and “ι” bend towards different directions and should be distinguished, while the tails in different poses should be treated the same. Also sadly, our design is so clumsy that it has not got the ability to tell which part is the tail. It simply builds some kernels and segments to capture certain feature blocks, which could be a tail, part of a tail, or a combination of tail and hip. We hope this part to be fixed with more training examples. Actually, in current version, we’ve added shadow nodes to the structure to deal with more complex situations like this. For tails in different directions, the shadow nodes could treat them as the same feature, while the directions in lower layers are still represented by different nodes. Yet “tails” are too difficult concept for this network, for now we use it to deal with hand-written number “1” leaning to different sides.
- It is recommended that the author increase the omniglot dataset to verify the generalization of the proposed algorithm.
It’s really a good point to add more convincing experiments. However due to my bad planning and the very low speed caused by the unmature design, the result is not accomplished yet. We’ll keep on experimenting.
- It is suggested that the author increase the number of training iterations and the accuracy effect diagram.
The network is only built/trained once without iterations that update parameters. We added this explanation in the paper and highlighted in yellow..
- It is recommended to cite important literature in the past three years, which are used in deep learning small target detection. such as:
--Hybrid Behrens-Fisher- and Gray Contrast–Based Feature Point Selection for Building Detection from Satellite Images
--Constraint-Based Evaluation of Map Images Generalized by Deep Learning
We've added YOLOv8(2023), [Unsupervised semantic image segmentation with mutual information maximization] (2021), YOLOv4 (2020) and [Constraint-Based Evaluation of Map Images Generalized by Deep Learning] (2022) to citations.
- How to solve the problem of low accuracy of segmentation boundary.
Improvements may be applied in calculating functions. Also, the covered areas could be more precise in future version which includes more training instances. Maybe in future work we could take advantage of super pixels to simplify the calculation and get better boundaries.
- It is recommended that the author further increase the efficiency experiments with other methods to verify the efficiency of the proposed algorithm.
We added the segmentation results of Siamese Mask R-CNN and an unsupervised segmentation network in Section 4. Results. We also added some segmentation result on failed detections to illustrate the properties and limitations of our design.
Round 2
Reviewer 1 Report
Comments and Suggestions for Authors
In terms of content, all of the comments from the first review were adequately addressed.
Some figures are not referenced in the text (16,18, 20, 21),
Some formulas are referenced as “formula (1)” and others as “formula 1”, not all are referenced (formula 7) or are referenced falsely (“formula x”)
Tables are not referenced.
Comments on the Quality of English Language
Overall, the English must be improved, most of the passages highlighted in the first review were not improved and the typos and errors still are in the paper. (See first Review PDF)
There are a lot of typos and grammatically the whole text must be improved.
Author Response
Dear Reviewer:
Thank you very much for your kind work and precious advices. We have added the references and the missing formula, which are highlighted in yellow. We also sent this manuscript for English editing. The revised manuscript is enclosed, please see the attachment. On behalf of my co-authors, we would like to express our great appreciation to your help and concern.
Yours Sincerely,
Jun Su
J_su@mail.sim.ac.cn

Reviewer 2 Report
Comments and Suggestions for Authors
The author has revised and improved according to the reviewers' comments, and it is recommended to publish.
Comments on the Quality of English LanguageMinor editing of English language required.
Author Response
Dear Reviewer:
Thank you very much for your kind work and approval for publishing. We have learnt a lot from your precious advices. In this review we added the references and the missing formula, which are highlighted in yellow. We also sent this manuscript for English editing. The revised manuscript is enclosed, please see the attachment. On behalf of my co-authors, we would like to express our great appreciation to your help and concern.
Yours Sincerely,
Jun Su
J_su@mail.sim.ac.cn
